

# Inter-rater and test-retest reliabilities of lumbar stiffness measurement in the postero-anterior direction using a portable algometer and the Kinovea program

Wantanee Yodchaisarn[1], Sunthorn Rungruangbaiyok[2], Maria de Lourdes Pereira[3] and Chadapa Rungruangbaiyok[1]

[1] Department of Physical Therapy, School of Allied Health Sciences, Movement Science and Exercise Research Center, Walailak University, Nakhon Si Thammarat, Thailand
[2] Department of Electrical Engineering, Faculty of Engineering, Rajamangala University of Technology Srivijaya, Songkhla, Thailand
[3] CICECO - Aveiro Institute of Materials & Department of Medical Sciences, University of Aveiro, Aveiro, Portugal

## ABSTRACT

**Background**. Back pain negatively impacts a person's quality of life and can cause major disability or even death. The measurement of spinal stiffness can be utilized as a promising tool to guide therapeutic decisions regarding physical therapy that result in effective back pain management. This study aimed to determine the reliability of instrumented postero-anterior (PA) stiffness assessment of the lumbar spine in asymptomatic participants by novice assessors using a portable algometer and the Kinovea program.

**Methods**. Thirty asymptomatic participants aged 18–25 years were enrolled in this study. Two novice assessors examined the participants for lumbar spinal stiffness at L1–L5 for two consecutive days. The algometer was applied to measure the PA force that applied to each lumbar. The stiffness assessment of each lumbar region was recorded as a video. The 600 data sets of assessment videos were imported into the Kinovea program to perform displacement measurements of each lumbar level. Spinal displacement values at 15 N were defined by graph plotting between force and displacement. The spinal stiffness values were defined by slope calculation. Both variables were analyzed for inter-rater and test-retest reliabilities using intra-class correlation coefficients (ICCs) and standard error of measurement (SEM). Bland-Altman analysis was applied to assess the inter-rater and test-retest systematical bias and limits of agreement of measuring displacement and stiffness.

**Results**. The inter-rater reliability of measuring the displacement and the stiffness of L1–L5 was moderate to good (displacement ICCs: 0.67–0.83, stiffness ICCs: 0.60–0.83). The test-retest reliability of measuring the displacement and stiffness of L1–L5 were moderate to good, ICCs: 0.57–0.86 and ICCs: 0.51–0.88, respectively. The inter-rater analysis's Bland-Altman plot showed that the systematic bias was 0.83 when measuring displacement and 0.20 when measuring stiffness and the bias of both parameters were in both directions. While the test-retest systematically biased measurements of

Corresponding author
Chadapa Rungruangbaiyok,
chadapa.bn@wu.ac.th

displacement and stiffness were −0.26 mm and 0.22 N/mm, respectively, and the bias of both parameters were in both directions.

**Conclusions**. The moderate-to-good inter-rater and test-retest reliabilities of the portable instrumented spinal stiffness assessment using a digital algometer and the Kinovea program by novice assessors were demonstrated in this study. Bland-Altman analysis showed that measuring stiffness was more stable and had less systematic bias than measuring displacement. To figure out how reliable the device is in general, more comprehensive studies should be comparatively conducted in the future on subgroups of patients with normal vertebra, hypomobile or hypermobile conditions.

# INTRODUCTION

Low back pain is a health problem related to the musculoskeletal system (*Andersson, 1999*; *Staal et al., 2003*). The most common musculoskeletal problem in today's society is thought to affect 60–80% of the world's population (*Global Burden of Disease Study 2013 Collaborators, 2015*). Back pain affects the quality of life of an individual and can lead to disability (*Shipp et al., 2009*; *Tella et al., 2013*). Therefore, an accurate diagnosis on the cause of mechanical back pain can lead to early and proper treatment. The spinal stiffness assessment can be used in clinical decisions related physical therapy by guiding intervention targets.

Physical therapists can use manual passive intervertebral movement (PIVM) in the back-to-front direction or postero-anterior (PA) direction to measure how stiff the spine is in people with low back pain. Spinal stiffness of the lumbar spine is one of the symptoms of low back pain, and it plays an important role in clinical decision-making and effective treatment (*Latimer et al., 1996a*; *Latimer et al., 1996b*). The physical therapist assesses the perception of resistance through the hands. However, it is a subjective assessment, and therefore the reliability of repeated measurements is relatively low (*Maher & Adams, 1994*; *Binkley, Stratford & Gill, 1995*). Consequently, the mechanical spinal stiffness measurement device was developed by creating an assessment tool to obtain quantitative results by measuring the force response and displacement of the spine at each level (*Latimer et al., 1996a*; *Latimer et al., 1996b*; *Owens et al., 2007*; *Tuttle, Barrett & Laakso, 2009*; *Wong et al., 2013*; *Hadizadeh, Kawchuk & Parent, 2019*). The relationship in the form of the curve between force and distance can be applied to indicate the segmental spinal stiffness (*Wong & Kawchuk, 2017*). The mechanical spinal stiffness measurement device has been shown good to excellent reliability indices (*Latimer et al., 1996b*; *Owens et al., 2007*; *Wong et al., 2013*; *Hadizadeh, Kawchuk & Parent, 2019*).

However, mechanical spinal stiffness measurement devices are not widespread due to their lack of portability, high cost, and lack of user friendliness (*Wong & Kawchuk, 2017*). Therefore, researchers have developed mechanically assisted spinal stiffness testing devices

(*Björnsdóttir et al., 2016*). The force, direction, or speed of the indentation must all be manually adjusted by the assessor (*Wong & Kawchuk, 2017*). Some researchers applied the devices available in hospitals or physiotherapy departments to assess the spinal stiffness of the spine (*Tuttle & Hazle, 2018*). They used an algometer, a force measurement device, and the Kinovea program, a motion analysis program, to measure spinal stiffness, which was done by assessors with advance clinical experience (*Tuttle & Hazle, 2018*). Therefore, it is inconclusive that the use of the mechanically assisted spinal stiffness-testing devices would be applicable to those with insufficient clinical experiences including physical therapy students or new graduate physical therapists. Furthermore, the parameters that indicated spinal stiffness remains unclear. There are two common parameters: spinal stiffness and displacement, which can be extracted from the force–displacement (F-D) curve. Therefore, the objective of this study was to determine the reliability of inter-rater and test-retest reliabilities of the spinal stiffness assessment by novice raters using a portable algometer and the Kinovea program in asymptomatic participants.

## MATERIALS & METHODS

### Study design

This cross-sectional study was conducted during September to December 2020. The test-retest and inter-rater reliabilities of lumbar stiffness were conducted in the laboratory room at the Physical Therapy Department, Walailak University, Nakhon Si Thammarat province, Thailand. During the testing period, the laboratory room's temperature, light, and noise were controlled. The study was approved by the Ethics Committee of Walailak University and followed the Declaration of Helsinki (approval number: WUEC-20-241-01).

### Participants

A total of 30 male and female students of Walailak University aged between 18 and 25 years with asymptomatic low back pain were recruited by a convenience sampling method. In order to identify rater dependence in spinal stiffness measurement and to rule out potential confounding factors of dysfunction or disease, healthy volunteers were then chosen to participate in the study. The exclusion criteria were: (1) pregnancy; (2) diagnosis of inflammation or a spinal infection; (3) history of spinal surgery; (4) spinal deformities such as scoliosis, (5) discomfort in the lower back, and (6) a percentage of body fat higher than the normal range.

### Sample size calculation

The sample size was estimated for intraclass correlation observed by two raters using the formula derived from *Walter et al. (1998)*. We estimated the statistical power of 80% by defining the expected reliability as 0.5, the significance level ($\alpha$) 0.05 (*Bujang & Baharum, 2017*). The sample size estimate indicated that at least 22 participants were required for each lumbar level.

### Instrumentations

In the current study, two instruments were used to verify the test-retest and inter-rater reliabilities of lumbar stiffness. A Commander digital algometer (the Commander™ console

and Algometer dynamometer), which can be applied the force from 0 to 110 N, was used to monitor the PA force on each lumbar segment. Previous studies demonstrated the high reliability of measuring the force (*Kinser, Sands & Stone, 2009*; *Reezigt et al., 2023*). The algometer and Commander console were factory calibrated and were automatically set to zero calibration when turned on. The camera, which was in a high resolution (12 million pixels), 4K @ 24/30/60 fps, 1080p @ 30/60/120/240 fps, HDR, and Dolby Vision HDR (up to 30 fps), was used to record the spinal stiffness while applying PA pressure *via* the algometer. The camera was set so the distance between the participant and the camera was 3 m, which could focus the lumbar region and algometer during the testing period. Finally, the Kinovea software version 0.9.5 (*Charmant, 2021*) which is a free 2-D motion analysis program and presented high reliability in measuring the distance (*Elwardany, El-Sayed & Ali, 2015*; *Puig-Diví et al., 2019*; *Fernández-González et al., 2020*), was applied to measure the displacement of all lumbar segments.

## Raters

Two novice raters were undergraduate physical therapy students. The raters were recruited from the senior undergraduate physical therapy students who passed the manipulation course and clinical practice under clinical instructors. The raters were female, 22 years old. All raters participated in a two-hour training for five sessions to be experience trained personnel on how to perform spinal stiffness measurements using a digital algometer. The training covered the use of the digital algometers on the lumbar region, palpating and marking the locations of the body parts, they also provided an explanation of the study's protocols to the pain-free spinal participants who were not included in the main study.

## Experimental protocol

All volunteers were provided detailed information about the study's objective and protocol. Participants were given informed consent before participating in the study and were assessed for exclusion criteria. They were also recorded demographic data, including gender, weight, height, percentage of body fat, underlying disease, and medicines.

The eligible participants were randomly allocated to Rater 1 or Rater 2. Rater 1 was assigned to determine the lumbar stiffness first if the participant drew Rater 1. On the first day, the participants were measured by two raters; the interval between the raters was 15 min or until the participant was allowed to be evaluated by the other rater. The next day, the participants were examined using the same protocol as the first day.

The two raters were assigned to apply PA pressure on each lumbar spine to determine the lumbar displacement. The participants lay on an electric adjustable bed in prone, arms at the sides. They exposed the cloth to the waist and attached the force display to the right wrist. The rater applied adhesive tape to fix the skin under the lumbar area, marking the five lumbar segments. The study was designed to randomize the sequence of the lumbar segment to determine the lumbar displacement. Then, each segment was applied PA pressure while participants used the algometer and held the pressure at the end range for 5 s. Each segment was determined only once, as the same protocol was used the next day. The measurement for each segment was recorded as a separate video clip. Two raters were

blinded to the results of each test. The assumption that spinal stiffness would typically not change over two days led to the choice to assess the individuals twice (*Björnsdóttir et al., 2016*). Due to the nature of the measurements, which rely on the instrument rather than memory, recall bias would be avoided. The rater stood beside the treatment bed assuring that their shoulders were vertically aligned with the algometer's handle, and then pushed down while the participant exhaled with increasing force until they reached the elastic barrier or end-feel.

## Data analysis

The recorded videos of spinal stiffness measurements performed on participants at five lumbar levels by 2 raters over 2 days were included in the data analysis. Video clips where the force display screen does not start at 0 N and the algometer shaft area is obscured were excluded. The Kinovea program version 0.9.5 was used to measure the displacement of each lumbar segment, as shown in Figs. 1 and 2. The first step was to import the recorded video into the Kinovea program. and then trim the video beginning at the force of 0 N with the algometer pad attached to the participant's skin. The line calibration was performed using the three cm length of the algometer's shaft, and then the line was defined as a vertical line. The following step was to define the tracking object; the area of interest was the junction between the algometer's handle and shaft. The tracking path was created automatically, frame by frame, and the marker could be adjusted as appropriate. The tracking path was ended at the end range of movement, when the algometer was removed from the participant's back. The relative vertical displacement parameter was selected as the displacement value, and then the displacement and force values for each segment were extracted and saved in an Excel file. The last step was to use the Excel program (Microsoft®, Redmond, WA, USA) to make a graph between force (N) and displacement (mm) (Fig. 3). Then, we had to figure out the displacement at a force of 15 N. We selected the force applied at 15 N because it covered the maximum force applied throughout every lumbar level. In order to reflect the resistance of the soft tissue before it enters the capsule and joint resistance, we calculated the stiffness from all regions of the F-D curve in this investigation, including the toe area. This was accomplished by computing the slope over small displacement ranges (every two mm), where it can be assumed to be linear, and then averaging the result to account for all stiffness. The slope was calculated by dividing the change in force by the change in displacement. The spinal displacement and stiffness served as outcome parameters for statistical analysis.

## Statistical analysis

Two-way mixed effects with absolute agreement and multiple raters and measurements of an Intraclass Correlation Coefficient (ICC 3, k) were calculated to determine the inter-rater and the test-retest reliabilities for displacement and stiffness measurements at each lumbar segment individually. We selected a two-way mixed-effects model because the selected raters were the only raters of interest (*Koo & Li, 2016*). We selected the type of ICC as "mean of k raters," where k means the number of raters, in this study conducted with two raters. For the definition selection, we selected absolute agreement for both inter-rater and
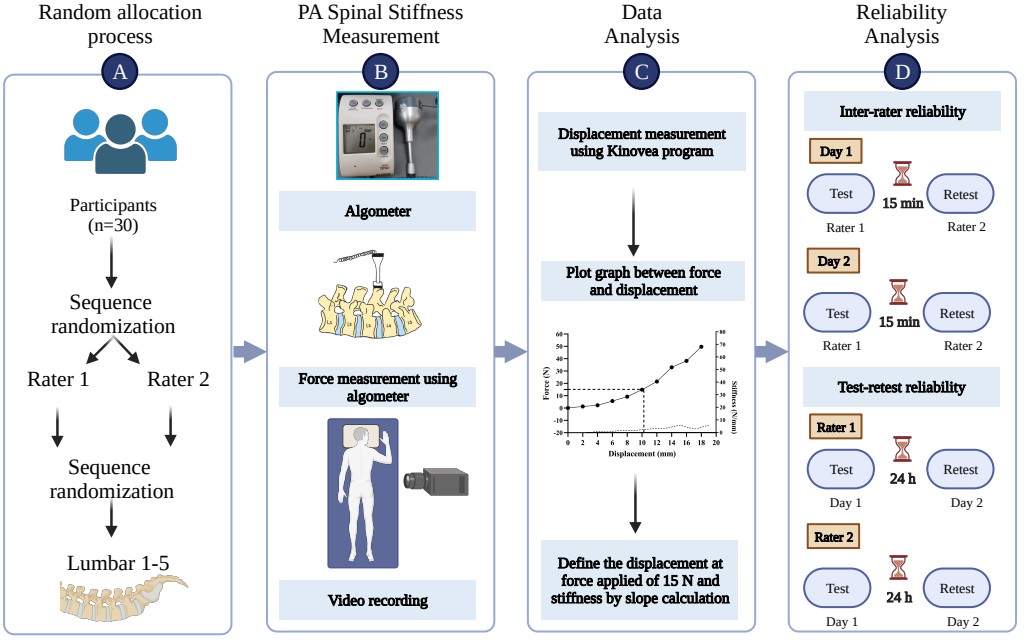

**Figure 1** **The illustration of methods in the study.** (A) Participant assignment; (B) postero-anterior spinal stiffness measurement; (C) data analysis; (D) reliability analysis.

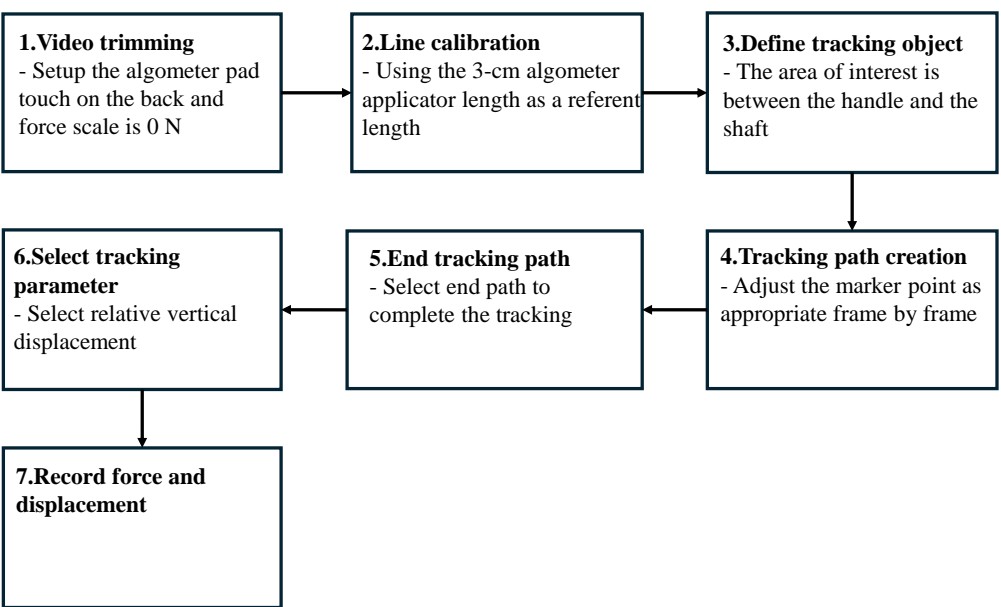

**Figure 2** **The illustration of displacement measurement protocol using Kinovea program.** 1, Video trimming; 2, line calibration; 3, define tracking object; 4, tracking path creation; 5, end tracking path; 6, select tracking parameter; 7, record force and displacement.

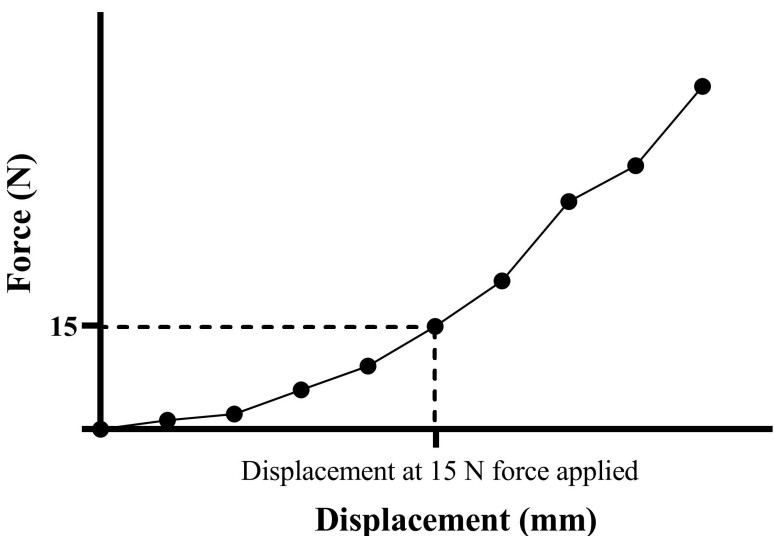

**Figure 3   Force–displacement curve demonstrating the displacement at 15 N force applied.**

test-retest reliabilities because we were concerned that two raters should measure the same score in the same subject for both days.

The standard error of measurement (SEM) was calculated to demonstrate the absolute reliability of the measurement using the following equation (*McManus, 2012*):

$$SEM = pooled\ SD \times \sqrt{(1 - ICC)}.$$

Furthermore, the Bland-Altman plot analysis was performed to determine the systematic bias and agreement range of spinal displacement and spinal stiffness measurements. The difference in spinal displacement and spinal stiffness values between raters (rater1–rater2) and between days (day 1–day 2) were plotted against the means of two raters and 2 days, respectively (*Giavarina, 2015*).

We used IBM SPSS Statistics version 25 (IBM, Armonk, NY, USA) to record, edit and enter all of our data for statistical analyses. The significance level was set at a *p*-value less than 0.05. The following criteria were used to evaluate the Intraclass Correlation Coefficient values: values less than 0.50 were considered poor, 0.50–0.75 were considered moderate, 0.75–0.90 were considered good, and 0.90–1.00 were considered excellent (*Koo & Li, 2016*).

## RESULTS

Thirty asymptomatic participants, 28 females and two males, with mean age of 21.46 (SD = 0.62) years, and mean body mass index of 20.11 (SD = 1.97) kg/m², were enrolled in this study.

No participant was excluded; however, 11 videos were excluded from the total of 600. Seven videos did not start at force application at 0 N, and two videos could not be performed line calibration due to obstructions at the algometer's shaft. Consequently, the sample size

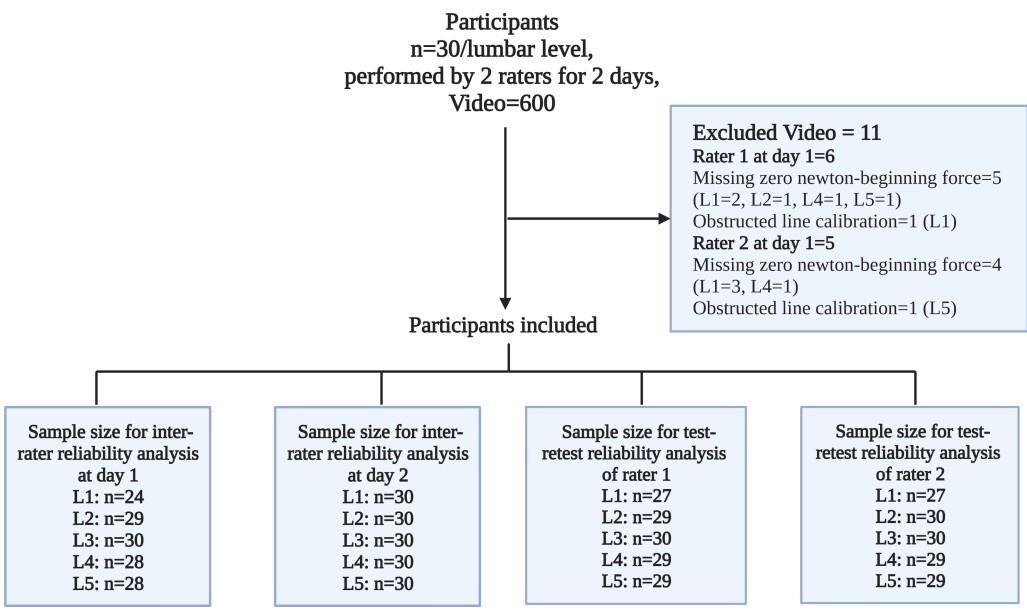

**Figure 4  Dataset inclusion and exclusion framework.**

for reliability analysis in each reliability category ranged from 24 to 30 participants, as demonstrated in Fig. 4.

Tables 1 and 2 show the reliability between raters and the reliability between tests for displacement at 15 N force and stiffness. The range of the maximum force that could be applied to each lumbar segment showed that the least amount of force that could be used was 16.20 N. The mean of displacement at 15 N of force applied ranged from 5.09 to 9.44 mm, and from 2.68 to 3.96 N/mm for stiffness. The results depend on the rater, day, and lumbar segment.

The inter-rater reliability analysis showed that the ICC ranged from 0.67 to 0.80 for measuring displacement on day 1 and from 0.60 to 0.83 for measuring stiffness. The ICCs of measuring displacement and stiffness on day 2 ranged from 0.68 to 0.83 and from 0.63 to 0.80, respectively. These results show moderate to good reliability of displacement and stiffness measurements between the two raters on both days. On day 1, the SEM for measuring displacement ranged from 1.18 to 1.80 mm and from 0.45 to 0.74 N/mm for measuring stiffness. On day 2, the SEM for measuring displacement ranged from 1.00 to 1.68 mm and from 0.47 to 0.78 N/mm for measuring stiffness.

The test-retest reliability analysis showed that the ICC for measuring rater 1's displacement ranged from 0.59 to 0.86, and the ICC for measuring stiffness ranged from 0.72 to 0.88. The ICCs for measuring displacement and stiffness for rater 2 ranged from 0.57 to 0.70 and from 0.51 to 0.63, respectively. These results show moderate to good reliability of displacement and stiffness measurements between the two days of rater 1, whereas measurements by rater had moderate reliability. The SEM for measuring the displacement of rater 1 ranged from 1.41 to 1.98 mm, and from 0.47 to 0.78 N/mm for

**Table 1 Inter-rater reliability.**

| Lumbar | Range of max. force (N) | Variables | Mean (SD) of variables | ICC (95% CI) | SEM |
|---|---|---|---|---|---|
| **At day 1** | | | | | |
| L1 (n = 24) | 16.20–61.60 | Displacement (mm) | 5.09 (2.31) | 0.67 (0.22–0.86) | 1.34 |
| | | Stiffness (N/mm) | 3.43 (1.16) | 0.60 (−0.14–0.85) | 0.74 |
| L2 (n = 29) | 18.00–68.60 | Displacement (mm) | 5.46 (2.39) | 0.76 (0.48–0.89) | 1.18 |
| | | Stiffness (N/mm) | 3.38 (1.05) | 0.70 (0.36–0.86) | 0.57 |
| L3 (n = 30) | 22.80–69.80 | Displacement (mm) | 6.33 (3.53) | 0.76 (0.48–0.89) | 1.74 |
| | | Stiffness (N/mm) | 3.43 (1.29) | 0.69 (0.33–0.85) | 0.72 |
| L4 (n = 28) | 24.60–73.40 | Displacement (mm) | 7.88 (3.97) | 0.80 (0.56–0.90) | 1.80 |
| | | Stiffness (N/mm) | 3.06 (1.10) | 0.82 (0.60–0.92) | 0.47 |
| L5 (n = 28) | 21.50–77.40 | Displacement (mm) | 8.36 (3.03) | 0.68 (0.27–0.85) | 1.72 |
| | | Stiffness (N/mm) | 3.09 (1.09) | 0.83 (0.64–0.92) | 0.45 |
| **At day 2** | | | | | |
| L1 (n = 30) | 17.10–42.60 | Displacement (mm) | 5.58 (2.38) | 0.83 (0.63–0.92) | 1.00 |
| | | Stiffness (N/mm) | 3.31 (1.35) | 0.67 (0.31–0.84) | 0.78 |
| L2 (n = 30) | 18.90–58.50 | Displacement (mm) | 5.62 (2.49) | 0.68 (0.32–0.85) | 1.41 |
| | | Stiffness (N/mm) | 3.26 (1.10) | 0.80 (0.58–0.91) | 0.49 |
| L3 (n = 30) | 21.10–51.90 | Displacement (mm) | 6.23 (3.12) | 0.79 (0.55–0.90) | 1.44 |
| | | Stiffness (N/mm) | 3.19 (1.21) | 0.79 (0.55–0.90) | 0.56 |
| L4 (n = 30) | 21.50–59.80 | Displacement (mm) | 7.78 (3.48) | 0.77 (0.46–0.90) | 1.67 |
| | | Stiffness (N/mm) | 2.94 (1.19) | 0.63 (0.23–0.82) | 0.73 |
| L5 (n = 30) | 22.00–55.00 | Displacement (mm) | 9.11 (3.00) | 0.69 (0.20–0.86) | 1.68 |
| | | Stiffness (N/mm) | 2.68 (0.99) | 0.78 (0.53–0.89) | 0.47 |

**Notes.**

Abbreviation: CI, confident interval; ICC, intra-class correlation coefficient; SD, standard deviation; SEM, standard error of measurement.

measuring stiffness. The SEM for measuring the displacement of rater 2 ranged from 1.42 to 1.99 mm and from 0.60 to 0.77 N/mm for measuring stiffness.

The Bland-Altman plot of the inter-rater analysis revealed the mean difference between raters for measuring displacement was 0.83 (95% CI [−4.34–6.00]) mm and 0.20 (95% CI [−1.90–2.31]) N/mm for measuring stiffness (Fig. 5). Figure 6 demonstrated the mean difference between days of measuring displacement was −0.26 (95% CI [−5.96–5.43]) mm and 0.22 (95% CI [−1.91–2.36]) N/mm for measuring stiffness.

## DISCUSSION

In this study, we investigated the inter-rater and test-retest reliabilities of spinal stiffness assessments of the lumbar spine in asymptomatic individuals adopting a portable algometer device and the Kinovea program conducted by novice assessors. In addition, we demonstrated two variables for representing the spinal stiffness, which were displacement at 15 N of force applied and average stiffness calculated by the slope of every 2 mm displacement change in the graph between force and displacement. We discovered moderate to good reliability for both measurements.

**Table 2  Test-retest reliability.**

| Lumbar | Range of max. force (N) | Variables | Mean (SD) of variables | ICC (95% CI) | SEM |
|--------|-------------------------|-----------|------------------------|--------------|-----|
| **Rater 1** | | | | | |
| L1 (n = 27) | 16.20–61.60 | Displacement (mm) | 5.31 (2.52) | 0.67 (0.28–0.85) | 1.45 |
| | | Stiffness (N/mm) | 3.69 (1.48) | 0.72 (0.38–0.87) | 0.78 |
| L2 (n = 29) | 19.30–68.60 | Displacement (mm) | 5.64 (2.31) | 0.59 (0.12–0.81) | 1.48 |
| | | Stiffness (N/mm) | 3.46 (1.04) | 0.74 (0.46–0.88) | 0.53 |
| L3 (n = 30) | 21.10–69.80 | Displacement (mm) | 6.84 (3.60) | 0.70 (0.36–0.86) | 1.98 |
| | | Stiffness (N/mm) | 3.38 (1.40) | 0.80 (0.58–0.90) | 0.62 |
| L4 (n = 29) | 21.50–73.40 | Displacement (mm) | 8.35 (3.79) | 0.86 (0.71–0.94) | 1.41 |
| | | Stiffness (N/mm) | 3.15 (1.34) | 0.88 (0.74–0.94) | 0.47 |
| L5 (n = 29) | 21.50–77.40 | Displacement (mm) | 9.44 (3.10) | 0.76 (0.50–0.89) | 1.48 |
| | | Stiffness (N/mm) | 2.85 (1.16) | 0.78 (0.52–0.90) | 0.54 |
| **Rater 2** | | | | | |
| L1 (n = 27) | 18.40–41.40 | Displacement (mm) | 5.58 (2.38) | 0.64 (0.23–0.84) | 1.42 |
| | | Stiffness (N/mm) | 2.95 (0.90) | 0.56 (0.03–0.80) | 0.60 |
| L2 (n = 30) | 18.00–53.60 | Displacement (mm) | 5.44 (2.56) | 0.59 (0.13–0.81) | 1.63 |
| | | Stiffness (N/mm) | 3.21 (1.09) | 0.60 (0.15–0.81) | 0.69 |
| L3 (n = 30) | 21.50–52.30 | Displacement (mm) | 5.72 (2.94) | 0.60 (0.15–0.81) | 1.85 |
| | | Stiffness (N/mm) | 3.25 (1.10) | 0.51 (−0.01–0.77) | 0.77 |
| L4 (n = 29) | 22.40–60.70 | Displacement (mm) | 7.28 (3.62) | 0.70 (0.36–0.86) | 1.99 |
| | | Stiffness (N/mm) | 2.91 (1.04) | 0.63 (0.21–0.82) | 0.64 |
| L5 (n = 29) | 21.50–60.20 | Displacement (mm) | 8.07 (2.98) | 0.57 (0.08–0.80) | 1.96 |
| | | Stiffness (N/mm) | 2.94 (0.96) | 0.53 (0.03–0.77) | 0.66 |

**Notes.**
Abbreviation: CI, confident interval; ICC, intra-class correlation coefficient; SD, standard deviation; SEM, standard error of measurement.

The present study's ICCs for inter-rater and test-retest reliability for measuring displacement were lower than those in the prior study; ICCs ranged from 0.83 to 0.97 (*Björnsdóttir et al., 2016*). In a previous study (*Björnsdóttir et al., 2016*), they measured the displacement of thoracic levels 6, 7, and 12 and lumbar level 1 at 8 kg of force application using the mechanically assisted spinal stiffness testing device. The force applied in a previous study was higher than in the current study. The current study demonstrated the maximum force application on L1 at 61.60 N. The use of a force at 80 newton is considered to be a use of force across the toe region and nearly the maximum range of force applied *in vivo* study (*Shirley, Ellis & Lee, 2002*), which took forces ranging from 10 to 90 N (*Wong et al., 2013*) and is comparable to giving grades III and IV mobilizations in Maitland's grading system (*Maitland, 2005*). While current study considered the displacement at force applied of 15 N, which was arranged in the early stages of movement. By this range of movement, it may still be in the toe-region range of motion where the force-orienting factor and force-generating speed may affect the reliability value. However, empirical studies should be conducted to further support this assumption. In addition, the mechanically assisted spinal stiffness testing device in a previous study was mounted into the displacement measurement (*Hadizadeh, Kawchuk & Parent, 2019*; *Owens et al., 2007*;

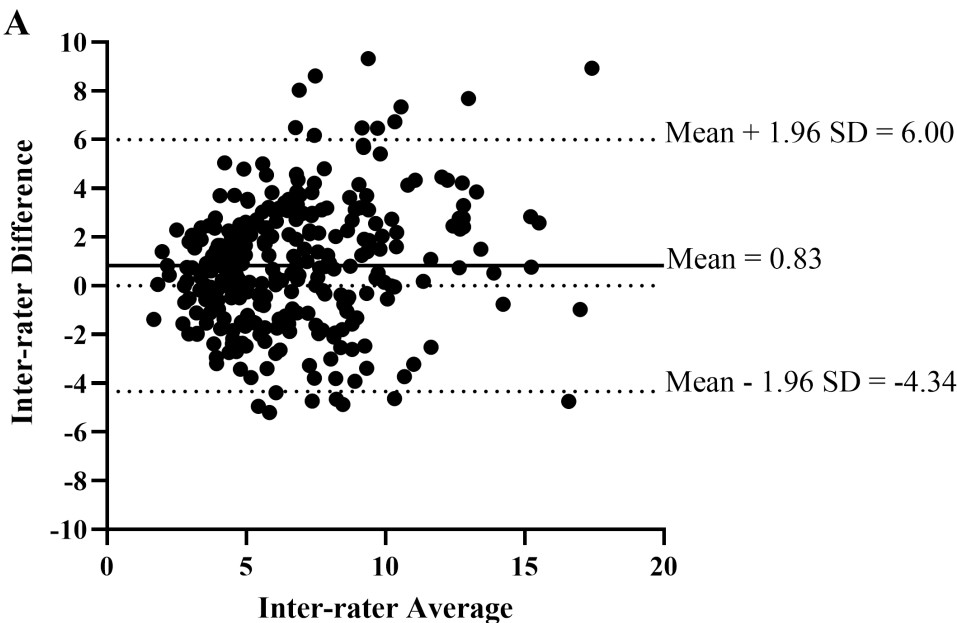

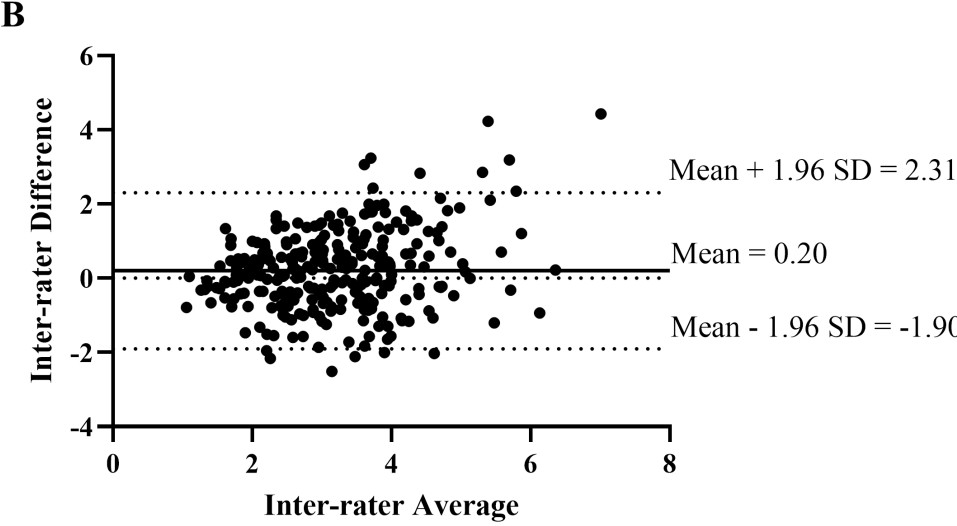

**Figure 5** The Bland-Altman plot of the inter-rater agreement of spinal displacement (A) and spinal stiffness (B) of L1–L5 between rater 1 and rater 2.

*Young et al., 2018*). Therefore, it is another factor that reduces the variation of reliability values. In the previous study (*Björnsdóttir et al., 2016*), the spinal stiffness was not reported for all of the spinal segments, and it was briefly said that the ICCs used to measure stiffness were worse than displacements. The ICCs of measuring stiffness for L1 were 0.48, while the ICCs of the current study were higher (0.64 to 0.83). In the previous study, the stiffness of the spine was determined by calculating the least-square cubic spline fit

**A**

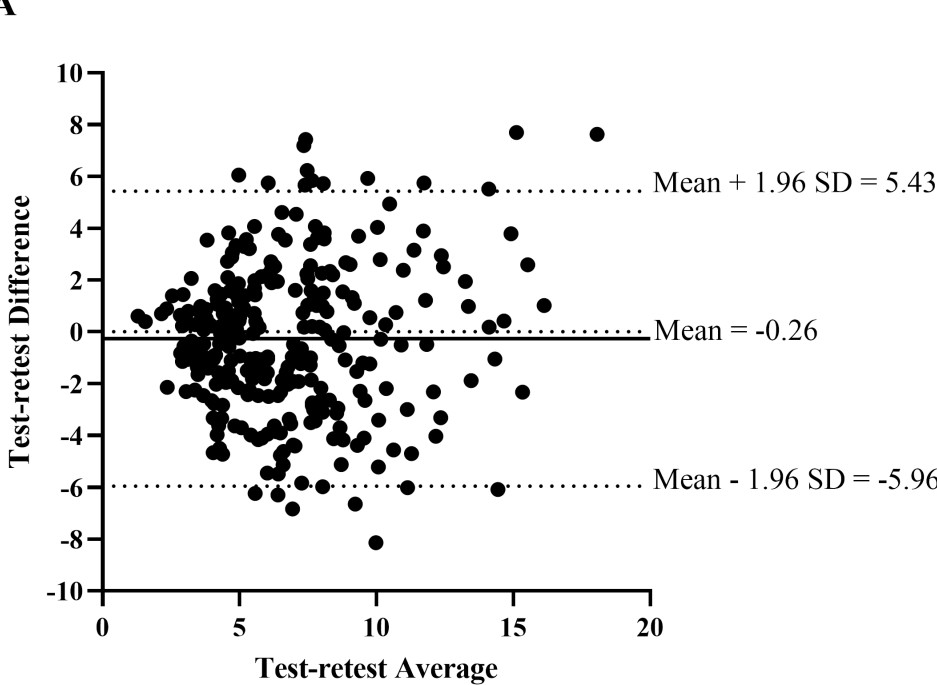

**B**

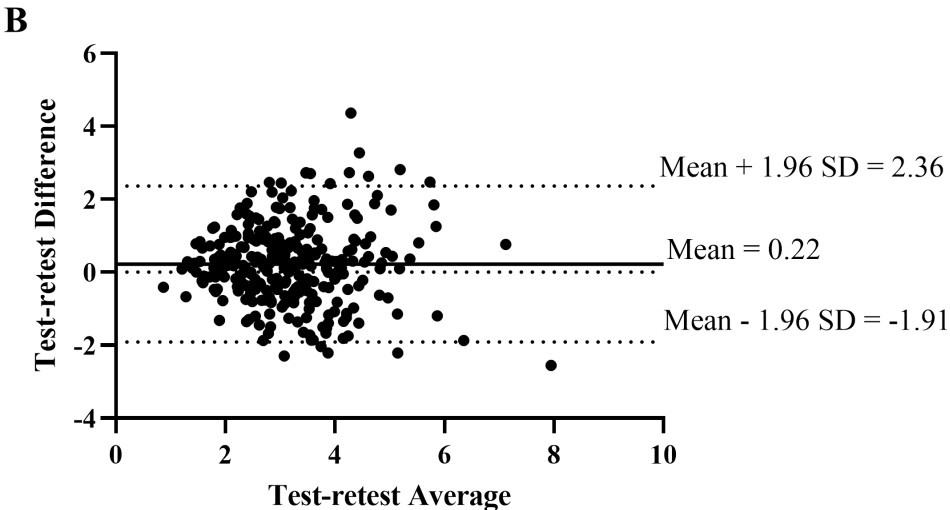

**Figure 6** The Bland-Altman plot of the test-retest agreement of spinal displacement (A) and spinal stiffness (B) of L1–L5 between day 1 and day 2.

(*Björnsdóttir et al., 2016*). The possible reason for the difference in reliability in stiffness depends on which part of the F-D curve was extracted for the stiffness. The toe region plays an important role for the stiffness value since it demonstrated the difference in the toe region in the F-D curve between hypomobility and normal vertebrae individuals (*Tuttle & Hazle, 2018*). Therefore, it should be counted as part of spinal stiffness. In Maitland's grading system, toe region could be shown by grades I to II of mobilizations, which means

that the resultant displacement took place without resistance (*Maitland, 2005*). In this study, we decided to get stiffness from all parts of the F-D curve, including the toe region, to reflect the resistance of the soft tissue before entering the capsule and joint resistance. We did this by calculating the slope in short ranges of displacement (every 2 mm), where the slope can be considered linear, and then averaging the value to represent all stiffness.

We also compared our test-retest ICCs for measuring the stiffness with the mechanical spinal stiffness measurement device; our ICCs (0.51 to 0.88) were lower than the previous study's (ICCs of 0.75 to 0.96) (*Hadizadeh, Kawchuk & Parent, 2019*). The main reason is that a mechanical spinal stiffness device can control the direction and speed of force application, which are major factors that disturb the reliability of measurements. However, the reliability of measurements by the methods in our setting is higher than that of manual segmental spinal stiffness assessment (*Maher & Adams, 1994*).

However, our study has some limitations. First, the mechanically assisted device cannot control the direction and speed of force application, which are major factors that might interfere with the reliability of measurements (*Wong & Kawchuk, 2017*). However, if the assessors are well trained to control these factors, it could further improve the reliability. Therefore, more future comprehensive study should be investigated to verify this hypothesis. Second, the participants in the study were only asymptomatic; normal vertebra or hypomobility condition was therefore not under initial screening. The previous study demonstrated the difference in the F-D curve between the hypomobile and the normal vertebra. The hypomobile vertebra's F-D curve showed less toe-region range (*Tuttle & Hazle, 2018*). Third, the 11 videos were excluded from the total of 600, consequently, the sample size of each lumbar level for both inter-rater and test-retest reliability ranged from 24 to 30 participants. The main reason is the lack of video recording rehearsal in the rater training protocol; therefore, future studies should include this step. Forth, the non-equal distribution of gender in this study (male less than female), it may not generalize equally to both sexes. Next, because the current study was time-consuming, it took 20 min for the force measurement protocol for five lumbar segments including the preparation period and 15 min for the data analysis per lumbar segment. It also took 20 min for single measurement, compared to prior study measuring spinal stiffness using mechanical device, it took 12 min for single measurement (*Young et al., 2018*). Lastly, due to the invasiveness and cost limitations, the measurement could not be included for validation; therefore, the gold standard measurements like fMRI or ultrasound diagnosis, is strongly recommended to validate the accuracy on the parameters of spinal stiffness measurement.

## CONCLUSIONS

The current study aimed to determine the inter-rater and test-retest reliabilities of mechanically assisted spinal stiffness testing devices using a portable algometer and the Kinovea program. The novice assessors performed the measurement at five lumbar segments in asymptomatic individuals. Although this study used novice assessors, it could also provide moderate to good reliability. The Bland-Altman analysis indicated that the inter-rater and test-retest measurements of stiffness illustrated less systematic bias and

were more stable than the measurements of displacement. Consequently, we suggest that our mechanically assisted spinal stiffness testing device (the algometer and the Kinovea program) is better suited for clinical than research applications, and the spinal stiffness variable is recommended for both inter-rater and test-retest measurements. To figure out how reliable the device is in general, more comprehensive studies should be comparatively conducted in the future on subgroups of patients with normal vertebra, hypomobile or hypermobile conditions.

## ACKNOWLEDGEMENTS

We express our gratitude to the participants for the contribution of their time, effort, and dedication. We would like to thank Associate Professor Doctor Manas Kotepui, School of Allied Health Science, Walailak University, for his guidance and suggestions for statistical analysis.

### Funding
The authors received no funding for this work.

### Competing Interests
The authors declare there are no competing interests.

### Author Contributions
- Wantanee Yodchaisarn conceived and designed the experiments, performed the experiments, authored or reviewed drafts of the article, and approved the final draft.
- Sunthorn Rungruangbaiyok analyzed the data, prepared figures and/or tables, and approved the final draft.
- Maria de Lourdes Pereira analyzed the data, authored or reviewed drafts of the article, and approved the final draft.
- Chadapa Rungruangbaiyok conceived and designed the experiments, performed the experiments, analyzed the data, prepared figures and/or tables, authored or reviewed drafts of the article, and approved the final draft.

### Human Ethics
The following information was supplied relating to ethical approvals (i.e., approving body and any reference numbers):

The Ethics Committee of Walailak University

### Data Availability
The raw measurements are available in the Supplementary File.

### Supplemental Information
Supplemental information for this article can be found online at http://dx.doi.org/10.7717/peerj.16148#supplemental-information.

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
