# Peer review of "Inter-rater and test-retest reliabilities of lumbar stiffness measurement in the postero-anterior direction using a portable algometer and the Kinovea program"

_PeerJ, doi:10.7717/peerj.16148_

## Round 0.1 · original submission · Major Revisions

Dear Authors

We have received the peer review and must inform you that your manuscript cannot be published in its current state, the reviewers have pointed out that your research requires major changes to be made in relation to methodological issues.

All the requirements suggested by the reviewers need to be addressed in great depth.

Reviewer 2 makes some important comments about the inconsistency of the results due to the limitations of the experimental design and the theoretical underpinnings of this research.

I recommend the authors in the introductory section to better justify the study by doing a thorough review of previous similar studies both with and without favourable results.

Once the authors have responded to the reviewers' requests, they should be ready to resubmit the manuscript.

·

Basic reporting

The scientific method is robust and valid. The language in the introduction setting up the importance of the research requires softening. Please see the review document attached with recommended changes.

Table 1 required a footnote detailing the abbreviations, and units reported.

Experimental design

I would appreciate greater detail about the protocols that novice assessors were trained in to assess stiffness. Especially due to 11 data points missing.

Validity of the findings

I require further elaboration re: the 11 missing data points. Perhaps just further clarification in the methods re: protocol.

Otherwise methodology, and analysis is appropriate.

Additional comments

The discussion was good for contextualising the findings from the study.

Reviewer 2 ·

Basic reporting

Thank you for giving me the opportunity to review this interesting manuscript.

The authors use a correct and understandable language for the reader.

Why displacement at 15 N or computing the slope change at every 2 mm of displacement? Authors should explain these assumptions here our move it on methods.

From the "coclusions" section, unappropriate paragraph formatting

The authors use numerous figures and tables. Some of them are redundant in the information: such as table 1. I suggest introducing these data in the text.
Figure 2 uses poorly defined images and screenshots. It is not possible to read its content.

Regarding raw data, why are more n/a on Displacement_L1 on D1? (familiarization progress?) If this affects the results you should explain and include on limitations

Experimental design

The authors indicate in the introduction that reliable assays already exist to measure the lumbar stiffness, but one of their limitations is that numerous steps are required to perform the measurement. After reading the methodology, and in particular the measurement protocol summarized in Figure 2, I note that the proposed protocol uses also numerous steps to obtain the data. The authors should justify in more detail why the protocol studied is simplified than those proposed by previous studies.


Please, you should indicate (on line 145) what is the range of the pressure (12, 15, 20 kg of pressure)?

Validity of the findings

The authors indicate that the sample is composed of men and women, but as indicated on line 239, there are 28 women. The data are not applicable to men (this should be reflected in the title).

Sample size calculation is not applied, 36% more participants were analyzed (which could lead to a significance bias per a bigger sample).

Conclusions are well stated, linked to original research question.


In summary, there are major limitations and inconsistencies associated with the design of this manuscript that can seriously influence the results and thereby the overall conclusions. Despite it is an interesting hypothesis, there are limitations that make me not recommend this article for publication:
1. The sample size calculation is not applied.
2. Non-homogeneity (sex variable) intra-group. The data are not applicable to men (this should be reflected in the title).
3. The hypothesis is justified because other tests have numerous steps (similar to this study) and need experienced therapists (the present protocol needs 10 hours of specific training to obtain moderate reliability). Regarding the raters, the reason for their assignment is not detailed and their descriptive variables are not adequate (for example, it is more interesting to know the academic report than the height of the raters).

---

## Round 0.2 · accepted · Accept

Dear Authors,

I am delighted to inform you that your manuscript has been accepted for publication in our journal. Congratulations on the hard work and dedication you've shown throughout the review process. We genuinely appreciate your submission and look forward to showcasing your contribution.

Warm regards,

Reviewer 2 ·

Basic reporting

The authors responded successfully to all suggestions performed. The manuscript, in spite of having several limitations (reflected in the article itself) can be considered, in my opinion, to be considered for publication.

Experimental design

The authors responded successfully to all suggestions performed. The manuscript, in spite of having several limitations (reflected in the article itself) can be considered, in my opinion, to be considered for publication.

Validity of the findings

The authors responded successfully to all suggestions performed. The manuscript, in spite of having several limitations (reflected in the article itself) can be considered, in my opinion, to be considered for publication.